# Evaluation of ATOX1 as a Potential Predictive Biomarker for Tetrathiomolybdate Treatment of Breast Cancer Patients with High Risk of Recurrence

**DOI:** 10.3390/biomedicines9121887

**Published:** 2021-12-12

**Authors:** Stéphanie Blockhuys, Camilla Hildesjö, Hans Olsson, Linda Vahdat, Pernilla Wittung-Stafshede

**Affiliations:** 1Department of Biology and Biological Engineering, Chalmers University of Technology, 412 96 Gothenburg, Sweden; pernilla.wittung@chalmers.se; 2Department of Clinical and Experimental Medicine, Linköping University, 581 85 Linköping, Sweden; Camilla.Hildesjo@regionostergotland.se (C.H.); Hans.Olsson@regionostergotland.se (H.O.); 3Memorial Sloan Kettering Cancer Center, New York, NY 10065, USA; vahdatl@mskcc.org

**Keywords:** breast cancer, ATOX1, copper depletion, tetrathiomolybdate, event-free survival, biomarker, clinical trial

## Abstract

Copper plays a key role in cancer metastasis, which is the most common cause of cancer death. Copper depletion treatment with tetrathiomolybdate (TM) improved disease-free survival in breast cancer patients with high risk of recurrence in a phase II clinical trial. Because the copper metallochaperone ATOX1 was recently reported to drive breast cancer cell migration and breast cancer migration is a critical factor in metastasis, we tested if ATOX1 expression levels in primary tumor tissue could predict the TM treatment outcome of breast cancer patients at high risk of recurrence. We performed ATOX1 immunohistochemical staining of breast tumor material (before TM treatment) of 47 patients enrolled in the phase II TM clinical trial and evaluated ATOX1 expression levels in relation with patient outcome after TM treatment. Our results show that higher ATOX1 levels in the tumor cell cytoplasm correlate with a trend towards better event-free survival after TM treatment for triple-negative breast cancer patients and patients at stage III of disease. In conclusion, ATOX1 may be a potential predictive biomarker for TM treatment of breast cancer patients at high risk of recurrence and should be tested in a larger cohort of patients.

## 1. Introduction

Copper levels are increased in serum and tumor tissue of cancer patients, and copper is required for at least three cancer hallmarks: proliferative immortality, angiogenesis and metastasis [1]. Since free copper is toxic, humans have evolved elaborate protein-based systems for uptake, intracellular transport, protein loading and storage of copper. The human copper proteome consists of at least 54 known copper-binding proteins (transporters, enzymes and many with unknown function) and expression levels of these proteins can be up- or downregulated in the different cancers in non-trivial patterns [2].

The copper metallochaperone antioxidant protein 1 (ATOX1) is upregulated in breast, colorectal, uterus and liver tumors, while downregulated in bile duct and pancreatic tumors [2]. ATOX1 regulates cellular copper homeostasis and transports cytosolic copper to the ATPases ATP7A and ATP7B in the trans-golgi network with further copper loading onto secretory enzymes, such as ceruloplasmin and lysyl oxidase [1]. Additionally, nuclear ATOX1 has been reported as a transcription factor and associated with activin A-stimulated colon cancer cell migration and colony formation [3]. In lung carcinoma cells, ATOX1 knockdown suppressed copper-stimulated cell proliferation [4]. In addition, we recently reported that ATOX1 stimulates directional breast cancer cell migration, a key step in metastasis, and that high tumor expression of ATOX1 significantly correlates with poor survival of breast cancer patients [5,6].

A recent phase II clinical trial tested the therapeutic efficacy of adjuvant tetrathiomolybdate (TM) following chemotherapy in triple-negative breast cancer (TNBC) patients, who showed no evidence of disease (NED), but were at high risk of recurrence. TNBC is characterized by its aggressive nature and lack of targeted treatment leading to a high prevalence of metastatic disease and death, respectively [7,8]. The phase II clinical TM trial results indicate an event-free survival (EFS) of 59.3% for TNBC patients at stage IV NED at a median follow-up of 10.3 years [9]. The results are striking considering that the mean survival rate for stage IV TNBC is 12 months [10].

Given that the reliance on copper trafficking in metastasis might vary across the different molecular subtypes of breast cancer and subsequently influence the impact of copper depletion as a therapeutic strategy, we sought to evaluate ATOX1 as a predictive factor for TM treatment outcome of breast cancer patients at high risk of recurrence using tumor tissue sections and follow-up data from the phase II clinical trial described above.

## 2. Materials and Methods

### 2.1. Patient Material

Patients with high risk of recurrent breast cancer were enrolled onto a phase II clinical TM study (NCT00195091) between 2007 and 2014. The study was approved by the Institutional Review Boards of Weill Cornell Medicine (IRB 0309006307) and Memorial Sloan Kettering Cancer Center (IRB 21-470). Additional details on the clinical trial design have been reported previously [9]. Briefly, after completing standard therapy, patients received oral daily TM for 2 years on the primary study. At completion they were eligible to enroll in a series of extension studies in 2 years increments. The primary endpoint was the effect of TM on bone-marrow-derived progenitor cells (VEGFR2+ endothelial progenitor cells). Secondary endpoints were outcomes for EFS and overall survival (OS), toxicity and multiple exploratory correlatives. At study entry, formalin-fixed paraffin-embedded tissue of primary tumor was obtained and archived. Of the 75 patients enrolled in the clinical study, 47 patients had adequate tumor material for evaluation of ATOX1 and were included in this study. Patient and treatment characteristics are shown in Appendix A, respectively.

### 2.2. ATOX1 Immunohistochemical Staining and Evaluation

The ATOX1 immunostaining of breast tumor tissue sections was performed as described previously [2]. Shortly, the paraffin-embedded tumor tissue sections were deparaffinized and underwent heat-induced epitope retrieval. After blocking endogenous peroxidase, the slides were incubated with the mouse monoclonal anti-ATOX1 primary antibody (H00000475-M01, Novus Biologicals, Littleton, CO, USA) followed by incubation with secondary antibody horseradish peroxidase (HRP)-Probe and tertiary antibody HRP-Polymer. For visualization, 3,3′-diaminobenzidine (DAB) chromogen in combination with DAB substrate buffer was applied to the tissue sections. Finally, the sections were counterstained with hematoxylin and mounted. The ATOX1 immunostaining was scored by two independent observers based on the staining intensity and the subcellular localization of positive staining without knowledge of clinicopathological and biological information. In cases of discrepant scoring results, a consensus score was reached after re-examination.

### 2.3. Statistics

Log-rank test was used to examine the relationship of ATOX1 staining with the relative probability for EFS. Survival curves were computed according to the Kaplan–Meier method in the SPSS software.

## 3. Results

### 3.1. ATOX1 Is Expressed in Both Nucleus and Cytoplasm of the Tumor Cells and Varies between Breast Tumor Tissue Sections of Patients in the Phase II Clinical Trial for TM

ATOX1 expression was evaluated in whole breast tumor tissue from breast cancer patients enrolled in the phase II clinical trial of TM using immunohistochemical staining. Visual inspection of the stained tumor sections indicated ATOX1 expression to vary between different tissue sections from negative to strong staining intensities and to be homogeneous within each tissue section, with comparable ATOX1 staining intensity in inner tumor and tumor border. A more detailed look showed ATOX1 staining in both nucleus and cytoplasm of different cell types, including epithelial cells, fibroblasts and immune cells.

We performed a semi-quantitative evaluation of ATOX1 staining intensities in the tumor cells only, considering both nucleus and cytoplasm, using intensity scores ranging from 0 to 3 representing negative, weak, intermediate and high staining intensities, respectively. In addition, we scored the percentage of tumor cells with the strongest ATOX1 staining intensity as a measure for tumor heterogeneity and found a high level of tumor homogeneity, with an average of 79% and 82% considering nuclear and cytoplasmatic ATOX1 levels, respectively. Illustrations for ATOX1 IHC staining and ATOX1 intensity scores are depicted in Figure 1A,B, respectively (Appendix A for positive and negative controls).

### 3.2. ATOX1 Expression Levels in the Tumor Cell Cytoplasm, and Not in the Tumor Cell Nuclei, Vary Depending on Subtype and Stage of Disease

For analysis, we divided the scoring data into two groups: low ATOX1, which includes negative and weak staining intensities (i.e., scores 0 and 1), and high ATOX1, which includes intermediate and strong staining intensities (i.e., scores 2 and 3). Figure 2 shows a swimmer plot with, for each patient, the clinical follow-up data (clinical response, TM treatment duration, subtype and stage of breast cancer disease) and ATOX1 expression levels in both tumor cell nucleus and cytoplasm.

ATOX1 intensity levels in the tumor cell nucleus and cytoplasm of whole tissue sections correlated for 44%, with 10 low and 31 high nuclear ATOX1 intensity cases versus 8 low and 39 high cytoplasmatic ATOX1 intensity cases. Upon comparison of the ATOX1 levels for different subtypes of disease, we found the highest prevalence of high cytoplasmatic ATOX1 levels in the tumor tissue of patients with HER2 subtype of disease (n_high_ = 16 out of 16 (100%)) followed by Luminal (n_high_ = 16 out of 18 (88%)) and TNBC subtypes (n_high_ = 17 out of 23 (74%)), and we found a similar prevalence of high nuclear ATOX1 levels in the tumor tissue from patients with Luminal, HER2 and TNBC subtypes of disease (n_high_ = 12 out of 18 (67%), n_high_ = 4 out of 6 (67%) and n_high_ = 15 out of 23 (65%), respectively) (Figure 3A,B). Comparison of the ATOX1 levels for different stages of disease showed higher prevalence of high cytoplasmatic ATOX1 levels for patients with disease stages III (n_high_ = 19 out of 23 (82.6%)) and IV (n_high_ = 19 out of 21 (90.5%)) versus stage II (n_high_ = 1 out of 3 (33.3%)) and a similar prevalence of high nuclear ATOX1 levels for patients with disease stages II, III and IV (n_high_ = 2 out of 3 (66%)), n_high_ = 16 out of 23 (69%) and n_high_ = 13 out of 21 (62%), respectively) (Figure 3C,D).

### 3.3. High ATOX1 Expression in the Tumor Cell Cytoplasm Correlates with a Better EFS upon TM Treatment Especially for TNBC Patients and Patients at Stage III of Disease

To understand if ATOX1 levels in the cancer tissue relate to TM treatment outcome, we evaluated the correlation between ATOX1 expression levels and EFS in the patients. We found that high ATOX1 levels in the tumor cell cytoplasm, but not ATOX1 in the nucleus, correlated with a trend towards better EFS when considering all breast cancer subtypes (i.e., Luminal, HER2 and TNBC) (Figure 4A,B).

Deeper investigation of the correlation between ATOX1 expression levels and patient outcome after TM treatment for the different subtypes separately demonstrated that this trend of better EFS for high cytoplasmatic ATOX1 was evident for the TNBC patient cohort, with an EFS of 82% for high ATOX1 versus 67% for low ATOX1 (*p* = 0.45) (Figure 5A). However, for Luminal and HER2 subtypes, the sample size, especially for low ATOX1 levels, was too low to draw conclusions. Evaluation of the correlation between cytoplasmatic ATOX1 levels and patient outcome after TM treatment for the different stages of disease demonstrated that the patients who were diagnosed at disease stage III, and not at disease stage IV, had a significantly better EFS when having high ATOX1 levels in their tumor. The EFS for stage III patients with high tumor cytoplasmatic ATOX1 was 81% versus 25% for patients with low ATOX1 (*p* = 0.018) (Figure 5B). There were too few cases of stage II patients to draw conclusions for this group (*n* = 3).

## 4. Discussion

Copper depletion with TM is a promising treatment for breast cancer patients. A first phase II clinical trial for TM treatment of breast cancer patients with high risk of recurrence demonstrated an EFS of 71.4% at a median follow-up of 10.3 years [9]. Since ATOX1 is a copper chaperone protein involved in breast cancer cell migration and breast cancer progression [5,6], we here evaluated ATOX1 as a predictive biomarker for the benefit of TM treatment using cancer tissue material and follow-up data from the breast cancer patients in the phase II clinical trial [9]. We specifically sought to explore whether there was an association between ATOX1 expression in the primary tumor (before TM treatment) and outcome in patients receiving TM as a therapeutic strategy.

Classical immunohistochemistry was used to manually score ATOX1 levels in tumor cells and the staining intensities were grouped into low (intensity scores 0 and 1) and high (intensity scores 3 and 4) ATOX1 levels. ATOX1 expression was found and evaluated for both nucleus and cytoplasm of the tumor cells. The staining results for ATOX1 in tumor tissue sections demonstrated that high ATOX1 levels for the tumor cell cytoplasm were most often found in the HER2 subtype, followed by Luminal and TNBC subtypes; in addition, high ATOX1 levels were more frequent at disease stages III and IV in comparison to stage II. These results are in agreement with our previous results of increased *ATOX1* mRNA expression levels for the HER2 subtype, followed by no change for the Luminal subtype and lower levels in the Basal subtype, and a trend of increasing *ATOX1* mRNA levels with higher stages of disease [6].

We found that breast cancer patients at stage III of disease with high cytoplasmatic ATOX1 levels in their primary tumor have a 65% higher EFS chance after copper depletion treatment with TM (median EFS is 40.4 months for low ATOX1 patients, while not yet reached for high ATOX1 patients). Evaluation of the different subtypes showed that TNBC patients with high cytoplasmatic ATOX1 levels had an EFS which is 15% higher than for patients with low cytoplasmatic ATOX1 levels. For the HER2 and Luminal subtypes, no conclusions were drawn due to the low sample size. The ATOX1 correlation in TNBC suggests that, here, ATOX1-dependent processes play a crucial role in cancer progression (such as metastasis), and therefore copper depletion with TM may have a beneficial effect [5,9,11]. Copper depletion with TM reduces circulating copper levels [9] and we hypothesize that consequently, cancer cells (with high demand for Cu) may become starved, while other cells remain unaffected.

Taken together, cytoplasmatic ATOX1 has the potential to be a predictive biomarker for TM treatment of breast cancer patients at high risk of recurrence, such that patients with high cytoplasmatic ATOX1 levels in the cancer tissue might benefit from TM treatment. We note that a larger sample size is urgently needed to come to conclusions for the HER2 and Luminal subtypes. However, the larger sample size for the TNBC subgroup implies that ATOX1 may be a predictive biomarker for TM in this cancer subtype. A larger randomized phase II clinical trial of TNBC patients is scheduled to begin in late 2021 supported by the Translational Breast Cancer Research Foundation and the NCI NExT program.

## Figures and Tables

**Figure 1 biomedicines-09-01887-f001:**
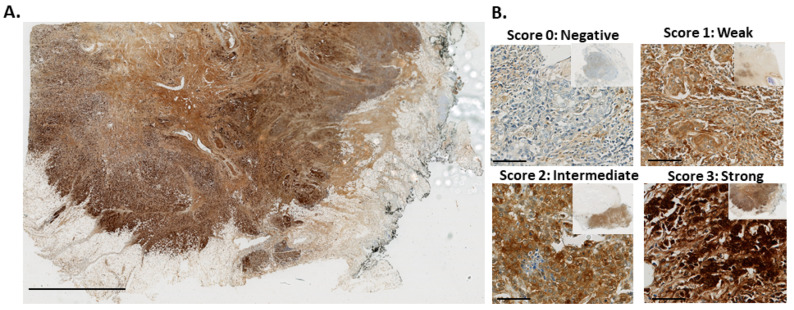
ATOX1 immunostaining of breast cancer tissue sections from patients in the phase II clinical trial for tetrathiomolybdate (TM) treatment. (**A**) ATOX1 immunohistochemical staining of a whole tumor tissue section (scale bar indicates 5 mm). (**B**) Examples of negative (score 0), weak (score 1), intermediate (score 2) and strong (score 3) ATOX1 staining intensities (scale bars indicate 100 µm).

**Figure 2 biomedicines-09-01887-f002:**
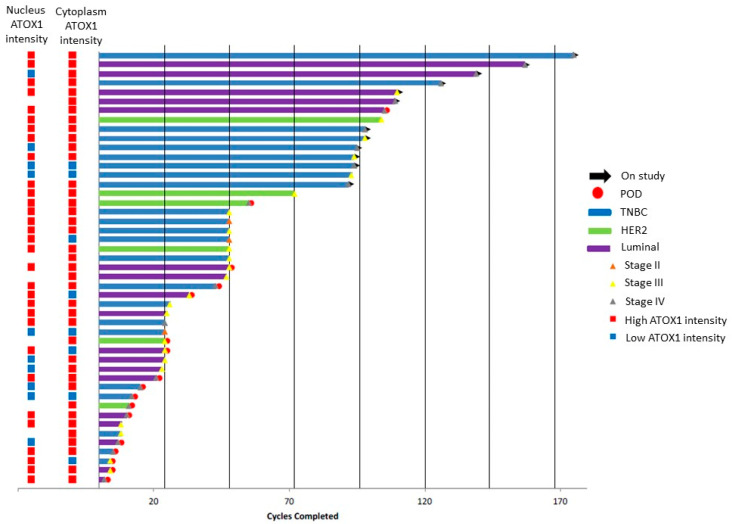
Swimmer plot representing the clinical trial cohort (n_patients_ = 47) with data on clinical response (POD = progression of disease), duration of therapy (in terms of TM treatment cycles), breast tumor subtype and stage of breast cancer for each patient, whereby each bar represents one patient in the study cohort. ATOX1 staining intensity levels in nucleus and cytoplasm are shown in the columns located to the left.

**Figure 3 biomedicines-09-01887-f003:**
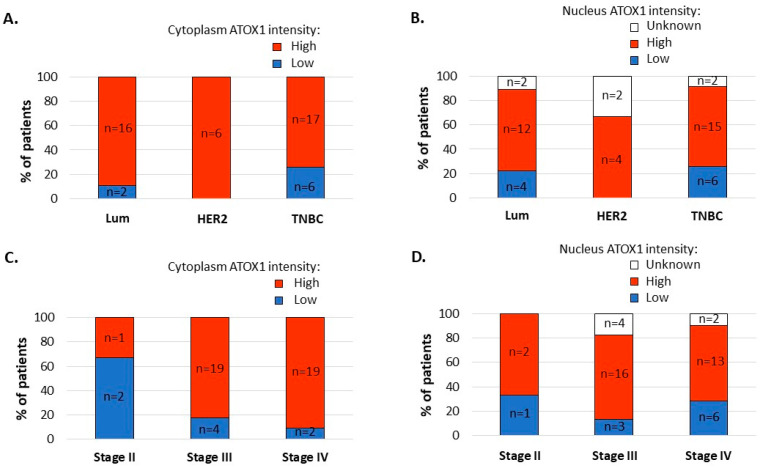
Graphical presentation of percentage of patients with low versus high ATOX1 intensity levels in the tumor cell cytoplasm or nucleus in the whole tumor tissue sections with differentiation between subtypes ((**A**,**B**), respectively) and stages of disease ((**C**,**D**), respectively). (‘Low ATOX1’ = negative + weak ATOX1 staining intensities and ‘High ATOX1’ = intermediate + strong ATOX1 staining intensities).

**Figure 4 biomedicines-09-01887-f004:**
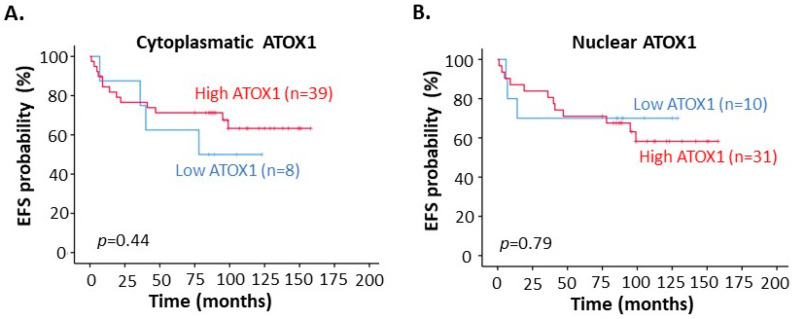
Kaplan–Meier curves for correlation between event-free survival (EFS) and ATOX1 intensity in (**A**) tumor cell cytoplasm and (**B**) in tumor cell nucleus of the breast tumor tissues from the phase II clinical trial for tetrathiomolybdate (TM) treatment. (‘Low ATOX1’ = negative and low ATOX1 staining intensities; ‘High ATOX1’ = intermediate and strong ATOX1 staining intensities).

**Figure 5 biomedicines-09-01887-f005:**
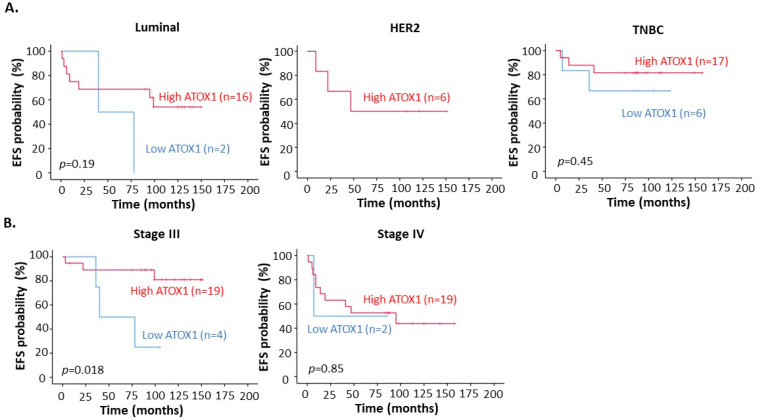
Kaplan–Meier curves for correlation between event-free survival (EFS) and ATOX1 intensity in tumor cell cytoplasm of whole breast tumor tissues with differentiation between (**A**) different subtypes and (**B**) stages of breast cancer disease.

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
