# Peer review of "Evaluation of ATOX1 as a Potential Predictive Biomarker for Tetrathiomolybdate Treatment of Breast Cancer Patients with High Risk of Recurrence"

_biomedicines, 2021, doi:10.3390/biomedicines9121887_

Round 1
Reviewer 1 Report
Congratulations for the paper.
The article is suitable for publication in the present form.
Author Response
Thank you very much for accepting our manuscript.
Reviewer 2 Report
1. Line 37, typo for heat
2. I Would request to arrange 3AC as 3AB and 3BD as 3CD for easy understanding.
3. Did the authors see some cases having both cytoplasimic and nuclear expression? It is evident in Fig.1A. The IHC image quality should be improved.
4. A better resolution Fig 4 should be added. Some months are not readable. p value is missing in Fig. 4B
5. More discussion on the expression pattern of ATOX1 and functional role in various cancer will be informative and useful for the readers.
6. A table with patients demographics will be helpful.
Author Response
Please find my response to your comments in the file attached.
Thank you for your comments.

Round 2
Reviewer 2 Report
The authors have addressed all concerns.